# Determining the sample size required to establish whether a medical device is non-inferior to an external benchmark

Adrian Sayers,[1,2] Michael J Crowther,[3] Andrew Judge,[4,5] Michael R Whitehouse,[1] Ashley W Blom[1]

[1]Muscloskeletal Research Unit, School of Clinical Sciences, University of Bristol, Bristol, UK
[2]School of Social and Community Medicine, University of Bristol, Bristol, UK
[3]Biostatistics Research Group, Department of Health Sciences, College of Medicine, Biological Sciences and Psychology, University of Leicester, Centre for Medicine, University Road, Leicester, UK
[4]NIHR Musculoskeletal Biomedical Research Unit, Nuffield Department of Orthopaedics, Rheumatology and Musculoskeletal Sciences, University of Oxford, Oxford, UK
[5]MRC Lifecourse Epidemiology Unit, University of Southampton, Southampton, UK

**Correspondence to**
Adrian Sayers;
adrian.sayers@bristol.ac.uk

## ABSTRACT

**Objectives** The use of benchmarks to assess the performance of implants such as those used in arthroplasty surgery is a widespread practice. It provides surgeons, patients and regulatory authorities with the reassurance that implants used are safe and effective. However, it is not currently clear how or how many implants should be statistically compared with a benchmark to assess whether or not that implant is superior, equivalent, non-inferior or inferior to the performance benchmark of interest. We aim to describe the methods and sample size required to conduct a one-sample non-inferiority study of a medical device for the purposes of benchmarking.

**Design** Simulation study.

**Setting** Simulation study of a national register of medical devices.

**Methods** We simulated data, with and without a non-informative competing risk, to represent an arthroplasty population and describe three methods of analysis (z-test, 1−Kaplan-Meier and competing risks) commonly used in surgical research.

**Primary outcome** We evaluate the performance of each method using power, bias, root-mean-square error, coverage and CI width.

**Results** 1−Kaplan-Meier provides an unbiased estimate of implant net failure, which can be used to assess if a surgical device is non-inferior to an external benchmark. Small non-inferiority margins require significantly more individuals to be at risk compared with current benchmarking standards.

**Conclusion** A non-inferiority testing paradigm provides a useful framework for determining if an implant meets the required performance defined by an external benchmark. Current contemporary benchmarking standards have limited power to detect non-inferiority, and substantially larger samples sizes, in excess of 3200 procedures, are required to achieve a power greater than 60%. It is clear when benchmarking implant performance, net failure estimated using 1−KM is preferential to crude failure estimated by competing risk models.

## Strengths and limitations of this study

► We propose a one-sample non-inferiority design for assessing the failure rate of medical devices against an external benchmark, using arthroplasty as an exemplar.
► Using a simulation study, we demonstrate that device failure rate estimated using Kaplan-Meier is appropriate and provides unbiased estimates of implant failure in the context of benchmarking arthroplasty.
► The number of individuals required at the beginning of a study in order to obtain nominal power at 5 different non-inferiority margins is described.
► The performance of three methods under two data generating processes is described in terms of bias, root-mean-square error, coverage and CI width.
► We assume that simple analyses using 1−Kaplan-Meier derived from population registry studies can provide causal interpretations.

is used to determine their efficacy. Without a head-to-head comparison against existing products, devices can be compared with external references or benchmarks. Product benchmarking in medical devices is common and is intended to help surgeons and healthcare administrators select safe and effective medical devices, yet there is no consensus on how this should be performed.

In arthroplasty, the process of benchmarking prosthetic implants is extensive. However, there is considerable debate with regards to the standards and criteria that should be adopted.[1] It has recently been suggested that the benchmark failure rate for prosthetic implants should become more stringent and reduced from the 10% to 5% at 10 years,[2] and the National Institute for Health and Care Excellence has incorporated this into guidance with regards to patients with end-stage arthritis of the hip.[3]

Benchmarking bodies around the world were quick to respond and the Orthopaedic Device Evaluation Panel (UK),[4] Prostheses

## INTRODUCTION

Arthroplasty prostheses are not currently required to undergo randomised clinical trials prior to their introduction into routine clinical practice, and postmarket surveillance

List Advisory Committee (Australia)[5] and Nederlandse Orthopaedische Vereniging (Netherlands)[6] bodies have set their highest benchmark to 5% failure at 10 years in total hip prostheses.[4–6] Despite subtle differences in each countries benchmarking system, a common question is how many implanted devices do we need to observe to determine if the device meets the specified benchmark or not? Both the UK and Netherlands require that there should be at least 500 implants remaining at risk at the end of the benchmarking period,[4 6] but with no clear rationale of why 500 implants are more or less suitable than 5000. Australia does not specify the numbers left at risk at the end of 10 years but requires data to be supported from their national arthroplasty registry,[5] which is one of the world's largest arthroplasty registers.

Despite the recommendations of a maximum failure rate of 5% at 10 years,[2] it is not described how the data should be compared with the proposed benchmark. Therefore, the evaluation process is somewhat ill defined, and uncertainty around the estimates is not formally included in the decision-making process. Furthermore, there is ambiguity about how failure should be estimated and whether or not net or crude failure (estimands) is of primary interest, with some authors recommending crude failure in implant survival studies.[7] Net failure is only possible in a hypothetical world of immortal patients, that is, the competing event (death) is prevented from occurring,[8] and crude failure is the real-world probability of failure allowing for the competing risk (CR) (death).[9]

While benchmarking in arthroplasty is used as an exemplar, similar problems and arguments can be made in any medical discipline that utilises medical devices or implants, for example, cardiac or cosmetic surgery.

## Hypothesis tests and power

In a simple setting of a clinical study with no loss to follow-up and no censoring, if implant survival is calculated and it is less than or equal to the benchmark, do we conclude that the implant has reached the benchmark? That is, $\widehat{P}_F <= P_{BM}$, where $\widehat{P}_F$ is the proportion of failed implants and $P_{BM}$ is the proportion of failure defined by the benchmark. It is clear that this statement does not include a description with regards to the uncertainty of the estimate. Therefore, as in the majority of the scientific literature, we would probably consider conducting a one-sample hypothesis test or (more preferably) constructing a CI of whether or not the proportion of implant failures at 10 years is not greater than the benchmark plus some level of uncertainty,[10 11] that is, $H_1 \widehat{P}_F \pm \phi(z_{1-\alpha/2}) \widehat{SE}(\widehat{P}_F) >= P_{BM}$, where $\phi_Z$ is a centile of a standard normal distribution, and is the type I error rate. An analytical strategy equivalent to this would allow us to make probability statements such as 'The average implant failure at 10 years is estimated to be $P_F$, and in repeated sampling, we would expect on $1 - \alpha/2\%$ of occasions that the true, unknown, value is contained within the CI.' This approach can be framed as a one-sample hypothesis test (z-test), of one proportion (see equation 1).

$$z = \frac{\widehat{P}_F - P_{BM}}{\sqrt{\dfrac{\widehat{P}_F\left(1 - \widehat{P}_F\right)}{n}}} \quad (1)$$

The temptation is to simply rearrange equation 1, solve the hypothesis test for n and conclude that this is the minimal sample size required to detect if an implant is superior to the benchmark. However, this would be incorrect as this fails to recognise the uncertainty introduced by sampling variability. With small samples, sampling variability is large, and conversely when samples are large, sampling variability is small. Sampling variability is one of the reasons that type I and type II statistical errors are made. A type I error is made when we incorrectly reject a valid null hypothesis, that is, that the true device failure rate at 10 years is equal to the benchmark, whereas a type II error is when the null hypothesis is incorrect, that is, the true failure rate is not equal to the benchmark, yet we fail to reject it. One minus the type II error rate is also known as power (see figure 1).

While there has been some debate with regards to the use of formal sample size calculations, which are determined on the basis of power and type I errors or by prespecifying the desired width of the CI,[12 13] we believe they are both useful tools for researchers and device manufacturers when planning clinical studies and determining the sample sizes required to draw reliable conclusions, by explicitly stating expected power of a sample and/or defining the desired width of estimated CI. When planning a clinical study, decisions need to be made with regards to the uncertainty in which the null hypothesis should be rejected (commonly but not necessarily $\alpha = 0.05$) and power ($1 - \beta$), which is the ability to correctly reject the null hypothesis when it is false. Therefore, simply solving a hypothesis test for $n$ fails to consider the power to detect an effect when it is truly present. A study with a sample size that gives 100% power will correctly reject the null hypothesis on all occasions (although in practice 100% power is likely impossible), and correspondingly, a sample size that gives 50% power will correctly reject the null hypothesis on only half of occasions. This is the essence of nearly all superiority designed clinical studies.

## Non-inferiority and benchmarking studies

However, in the context of a benchmarking system, the definition of superiority is restrictive. For example, if the true failure rate of the implant of interest is exactly equal to the benchmark ($\widehat{P}_F = P_{BM}$), it would be desirable to say that the implant has met the required standard. Furthermore, it is impossible to demonstrate that it is superior because SE will always be positive and not 0, and therefore, the upper CI will always be greater than the benchmark ($(\widehat{P}_F + \Phi(z_{1-\alpha/2}) \widehat{SE}(\widehat{P}_F) > P_{BM}$). Therefore, in a superiority study design, if a prosthesis failure rate is truly equivalent to the benchmark (the null hypothesis is correct), in repeated sampling, we should only incorrectly reject it on 5% of occasions if $\alpha = 0.05$. Therefore, when comparing

The truth

|  | | $H_0$ is True | $H_0$ is False |
|---|---|---|---|
| Decision with observed data | Reject $H_0$: | Type I Error $(\alpha)$ | True positives or power $(1 - \beta)$ |
|  | Fail to reject $H_0$: | True negatives | Type II Error $(\beta)$ |

$H_0$= Null hypothesis, $H_1$= alternate hypothesis

$H_0$ & $H_1$ in a non-inferiority benchmarking setting.

$H_0$: Device failure rate is greater than the benchmark

$H_1$: Device failure rate is less than or equal to the benchmark

**Figure 1**  Simple hypothesis testing framework and illustration of type I and II errors, power and true negatives.

devices to a benchmark, a non-inferiority study design including sample size calculations, analysis and reporting maybe preferable.[14–18]

Despite the linguistic similarities between superiority and non-inferiority studies, the analysis and interpretation are different. A non-inferiority framework requires the interested parties to place limits around what could be described as non-inferior, that is, a non-inferiority margin ($\delta$). This is stating that, if a device failure rate was 5%, the 95% CI ranged between 4.01% and 5.99% and that the non-inferiority margin was 1%, we would be happy to conclude that device was clinically equivalent or non-inferior (see figure 2).

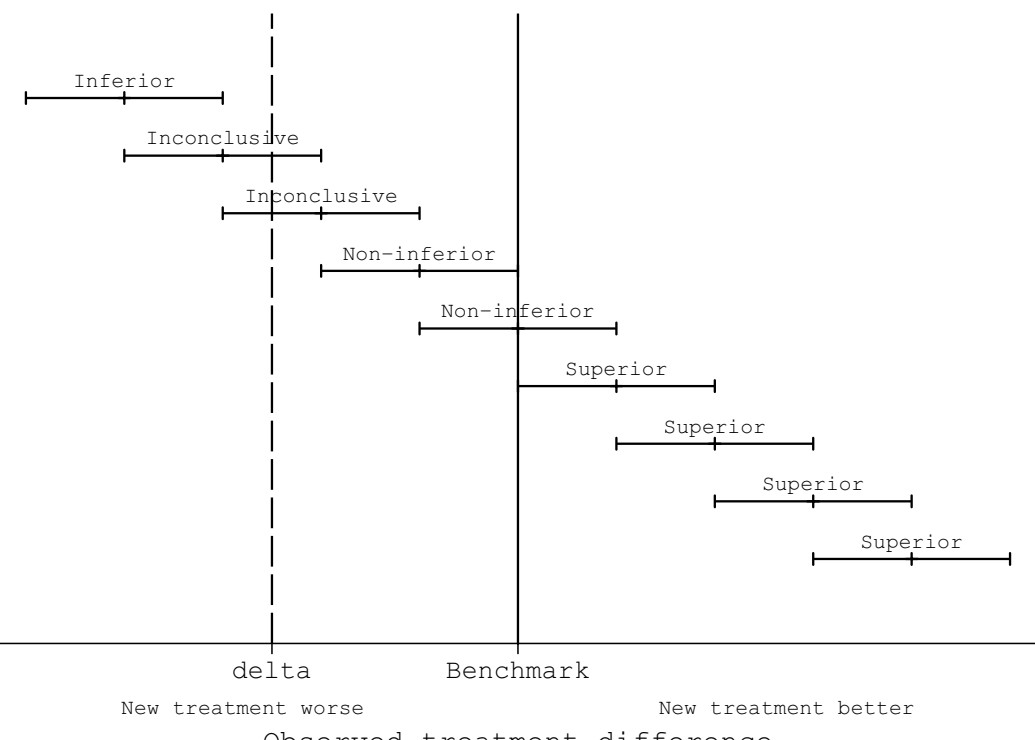

**Figure 2**  Schematic representation of inferiority, non-inferiority and superiority studies.

If the failure rate was 5.5% and the CI ranged between 5.25% and 5.75%, we would still conclude that this device was clinically non-inferior, despite being statistically inferior, with a non-inferiority margin of 1%. The methods by which one should choose an appropriate non-inferiority margin are inherently subjective, and the risk of choosing too large a margin represents a risk of exposing patients to inferior and less efficacious products. This is opposed to a margin that is too small, which in turn limits products of similar performance being introduced to the market; both the Food Drug Administration[19] (USA) and European Medicines Agency[20] provide extensive guidance in relation to the choice of non-inferiority margins.

## Aims

The aim of this study is to investigate the sample size required at the beginning of a study to demonstrate superiority and non-inferiority of implant failure compared with an external benchmark level of performance, that is, a one-sample non-inferiority study design in the presence of censoring, and the consequences of using three common methods of estimating failure in a simulation study.

## METHOD

The simulation study will be described using an Aims, Data generating process, Method of analysis, Estimands and Performance structure.[21]

## Aims

The aim of this study is to describe the sample size required to identify if a prosthetic implant has a failure rate non-inferior to an external benchmark using simple analytical solutions. In addition, we will use a simulation study to determine the power to detect superiority and non-inferiority with different sample sizes, different estimands, in the presence of a non-informative CR (mortality) and when the true implant failure rate is the same as the benchmark.

### Analytical sample size calculation of one proportion

Using analytical solutions of a z-test in a non-inferiority setting (see equations 2 and 3), the sample size at 5 non-inferiority margins ($\delta$) at 1, 2, …., 5% was calculated with power ($1 - \beta$) varied from 10% to 99.9%. The proportion of failures ($\widehat{P}_F$) was assumed to be equal to the benchmark ($P_{BM}$), 5%, and the type I error rate was set at $\alpha = 2.5\%$ (the one-tail equivalent of $\alpha=0.05$).

$$z = \frac{\widehat{P}_F - P_{BM} - \delta}{\sqrt{\frac{\widehat{P}_F\left(1 - \widehat{P}_F\right)}{n}}} \tag{2}$$

$$n = \widehat{P}_F\left(1 - \widehat{P}_F\right)\left(\frac{z_{1-\alpha} + z_{1-\beta}}{\widehat{P}_F - P_{BM} - \delta}\right)^2 \tag{3}$$

## Data Generating Process

Two different data generating mechanisms with varying sample sizes (n=100, 200, 400, 800, 1600, 3200, 6400) were explored: (DGP1) implant failure with no censoring and (DGP2) implant failure with censoring for mortality. Implant failure and mortality data were simulated independently (non-informative censoring) from a parametric survival distribution (2-parameter Weibull) (failure: $T \sim Weib\left(\lambda = 0.01, \gamma = 0.71\right)$; mortality: $T \sim Weib\left(\lambda = 0.017, \gamma = 1.32\right)$, where $\lambda$ and $\gamma$ are scale and shape parameters in a Weibull distribution, respectively.[22 23] Parameters were chosen so failure and mortality were simulated to have 5% and 30% occurrence at 10 years, respectively, which is similar to data from the National Joint Registry of England Wales and Northern Ireland,[24] and administratively censored at 10 years. A total of 1000 repetitions of each data generating process (DGP) were simulated; assuming nominal coverage of a z-test was achieved, Monte Carlo error would be equal to 0.7%.

## Method of analysis

We investigated the first DGP (no CRs) using the z-test (see equation 2) where the proportion of failures is simply estimated by the number of observed failures (f) within the sample divided by the total sample.

$$\widehat{P}_F = \frac{f}{n} \tag{4}$$

A failure function, $1 - \widehat{S}(t)$[25] (1−KM), approach was used to estimate failure where $n_j$ is the number at risk of failure preceding time $t_j$ and $f_j$ is the number of failures at time $t_j$ (see equation 5), with CIs estimated using asymptotic variance of $log\left(-log\widehat{S}\left(t_j\right)\right)$.[26]

$$\widehat{S}(t) = \prod_{j|t_j \leq t} \left(\frac{n_j - f_j}{n_j}\right) \tag{5}$$

The second DGP was similarly investigated using three approaches: (1) a z-test, where the proportion of failures was calculated excluding those who died prior to 10 years; (2) a failure function, $1 - \widehat{S}(t)$[25] (1−KM), with the addition of censoring individuals when they died, was also used; (3) a non-parametric CR model[27] was also explored (see equation 6). The CR model calculates the cumulative incidence for failure type k. It is estimated using a KM approach for all failure types at the instance before the failure of interest, $\widehat{S}(t_{j-1})$, and then multiplying it by the cause specific hazard, $d_{kj}/n_j$, that is, those experiencing the failure of interest divided by all individuals who remain at risk.

$$\widehat{CIF}_k(t) = \sum_{j|t_j \leq t} \widehat{S}(t_{j-1})\frac{d_{kj}}{n_j} \tag{6}$$

All analyses were conducted in Stata (Stata Statistical Software: Release 14.1).

### Estimands

The estimand of interest was cumulative implant failure at 10 years and its 95% CI in a hypothetical world where implant failure is the only possible outcome.

The first DGP explicitly simulates data where implant failure is the only possibly outcome, and the analytical methods purport to estimate net failure, whereas the second DGP simulates data where there are two potential outcomes (implant failure or death). The simple proportion of failures ($\hat{P}_F$) and failure function ($1-\hat{s}(t)$) both estimate net failure (assuming the untestable assumption of independence between CR and implant failure), whereas the CR approach ($\widehat{CIF}_k(t)$) estimates crude failure, which can be described as the real-world probability of failure, which allows for the fact that some individuals die before their implant fails.

### Performance

Performance was assessed in the superiority study setting using bias, $\frac{1}{n}\sum_{i=1}^{n}\left(\hat{P}_{F.i} - P_{BM}\right)$, root-mean-square error (RMSE) (absolute error), $\frac{1}{n}\sum_{i=1}^{n}\sqrt{\left(\hat{P}_{F.i} - P_{BM}\right)^2}$, power to detect superiority, that is, a non-inferiority margin at 0, which is equal to type I error in a one-tailed test, $\frac{1}{n}\sum_{i=1}^{n}100\left(\hat{P}_{F.upp.i} \leq P_{BM}\right)$, coverage of the 95% CI, $\frac{1}{n}\sum_{i=1}^{n}100\left(\hat{P}_{F.low.i} \leq P_{BM} \leq \hat{P}_{F.upp.i}\right)$, and CI width, $\frac{1}{n}\sum_{i=1}^{n}\left(\hat{P}_{F.upp} - \hat{P}_{F.low}\right)$. Similarly, we estimated performance in a non-inferiority setting using power (1−type II error) to detect non-inferiority at 1%, 2%, 3%, 4% and 5% at 10 years, $\frac{1}{n}\sum_{i=1}^{n}100(\hat{P}_{F.upp.i} \leq P_{BM} + \delta)$.

## RESULTS

### Analytical approach to sample size calculation of non-inferiority against a benchmark

Using analytic sample size calculations, non-inferiority against a benchmark sample size was calculated and is presented in figure 3.

With a non-inferiority margin of 3% failure, power of 50%, 203 individuals are required at the beginning of the study, whereas with 90% power, 555 individuals are required at the beginning of the study. There is an approximately log-linear association between sample size and power between 50% and 90%, at all non-inferiority margins. However, sample size rapidly increases as the non-inferiority margin reduces.

### Simulation approach to sample size calculation of superiority against a benchmark

Results from the simulation study using a superiority design are presented in figure 4.

The method of analysis and DGP process are indicated using five different coloured line styles. It is clear that the cumulative incidence function (CIF), which estimates crude failure, estimated in the presence of CR consistently underestimates net failure by 0.5%, whereas the z-test, an estimate of net failure, in

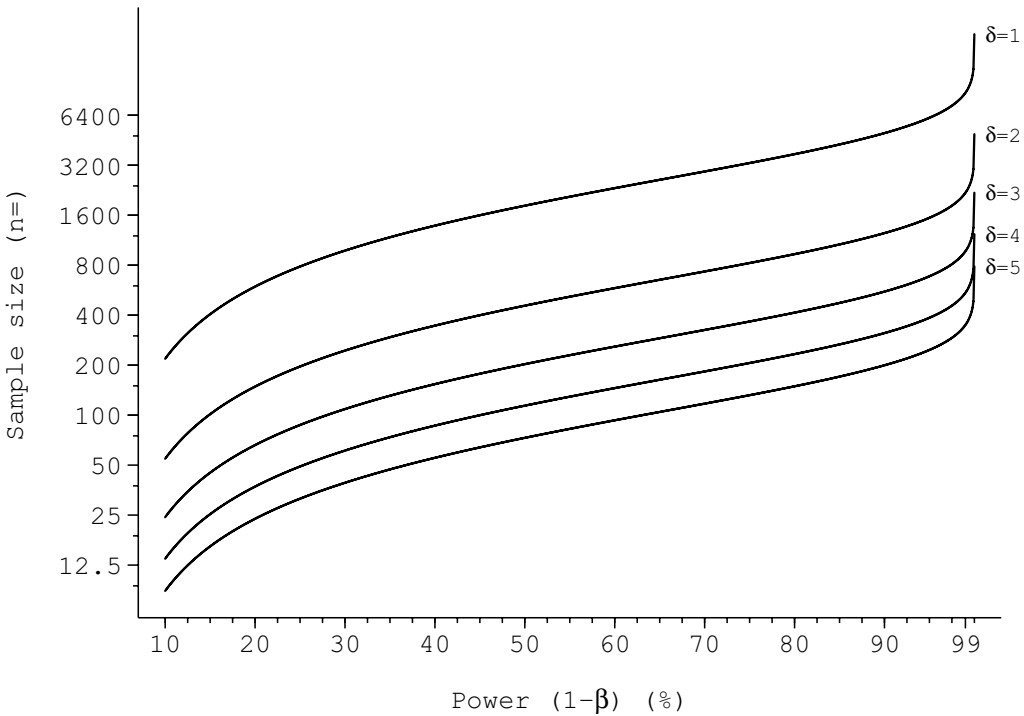

**Figure 3** The sample size required to detect non-inferiority of the failure proportion with non-inferiority margins (δ) at 1%, 2%, 3%, 4% and 5%.

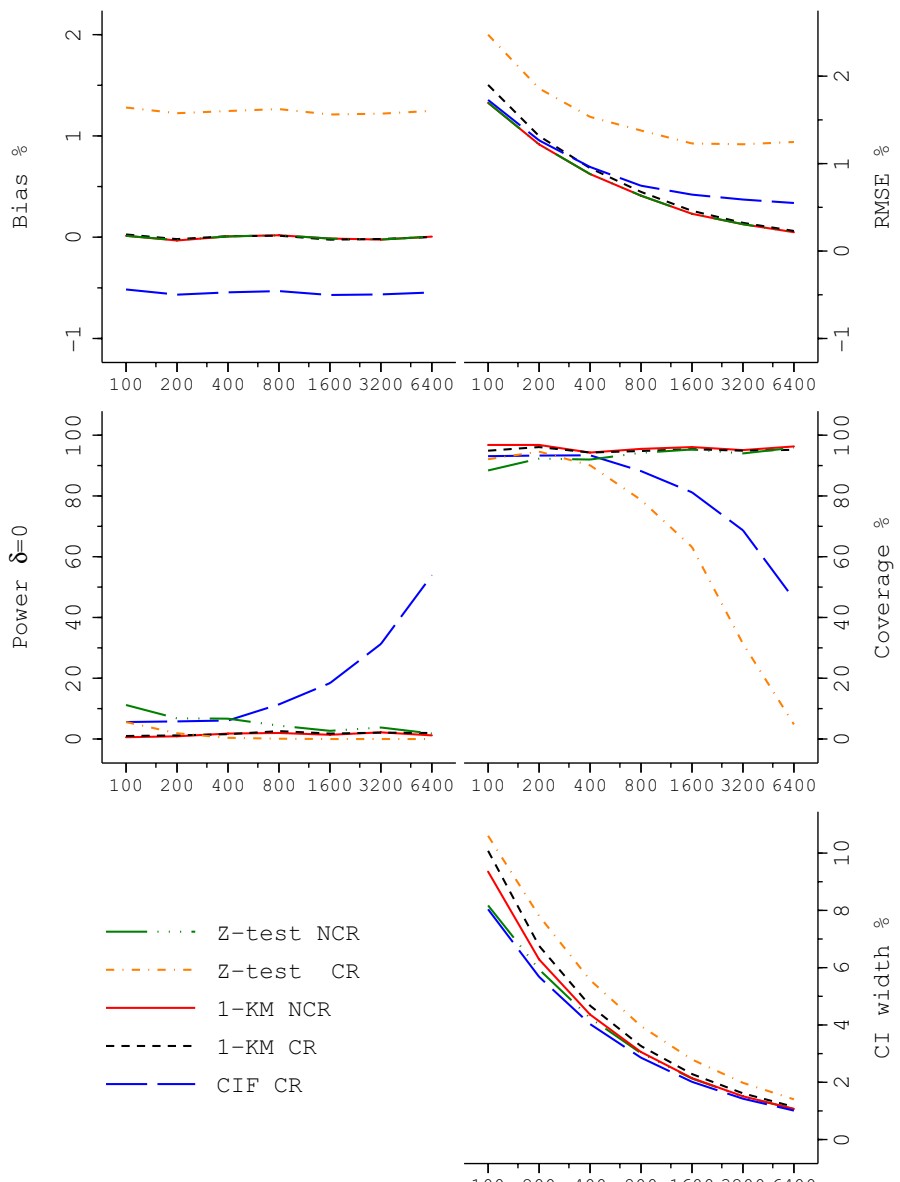

**Figure 4** Performance characteristics of five analyses; z-test and Kaplan-Meier when there is no competing risk (NCR) and a z-test, KM and cumulative incidence function (CIF) in the presence of competing risk (CR).

the presence of CR overestimates failure rates. Notably, the accuracies (RMSE) in the estimates of all methods are similar, and as sample size increases, they reduce. However, CIF and z-test in the presence of CR are biased estimates of net failure in comparison to 1–KM, and RMSE does not tend to 0 with increasing sample sizes. Correspondingly, coverage of the 95% CI reduces as the sample size increases for both CIF and z-test. The CIF power to detect superiority erroneously increases as a consequence of a consistent difference in the estimand (crude vs net failure) and narrowing CIs. The width of the estimated CIs, across all methods, consistently decreases as sample size increases. Despite their

homogeneity, the width of CI from CIF is approximately 1% larger than that of the 1–KM estimate in the presence of CR.

### Simulation approach to sample size calculation of non-inferiority against a benchmark

Simulation results using a z-test with no CRs were compared with analytic sample size calculations assuming no CRs (see online supplementary figure 1). Simulation results are generally concordant with analytical results. However, analytical results tend to slightly overestimate power when the non-inferiority margin is greater than 3%.

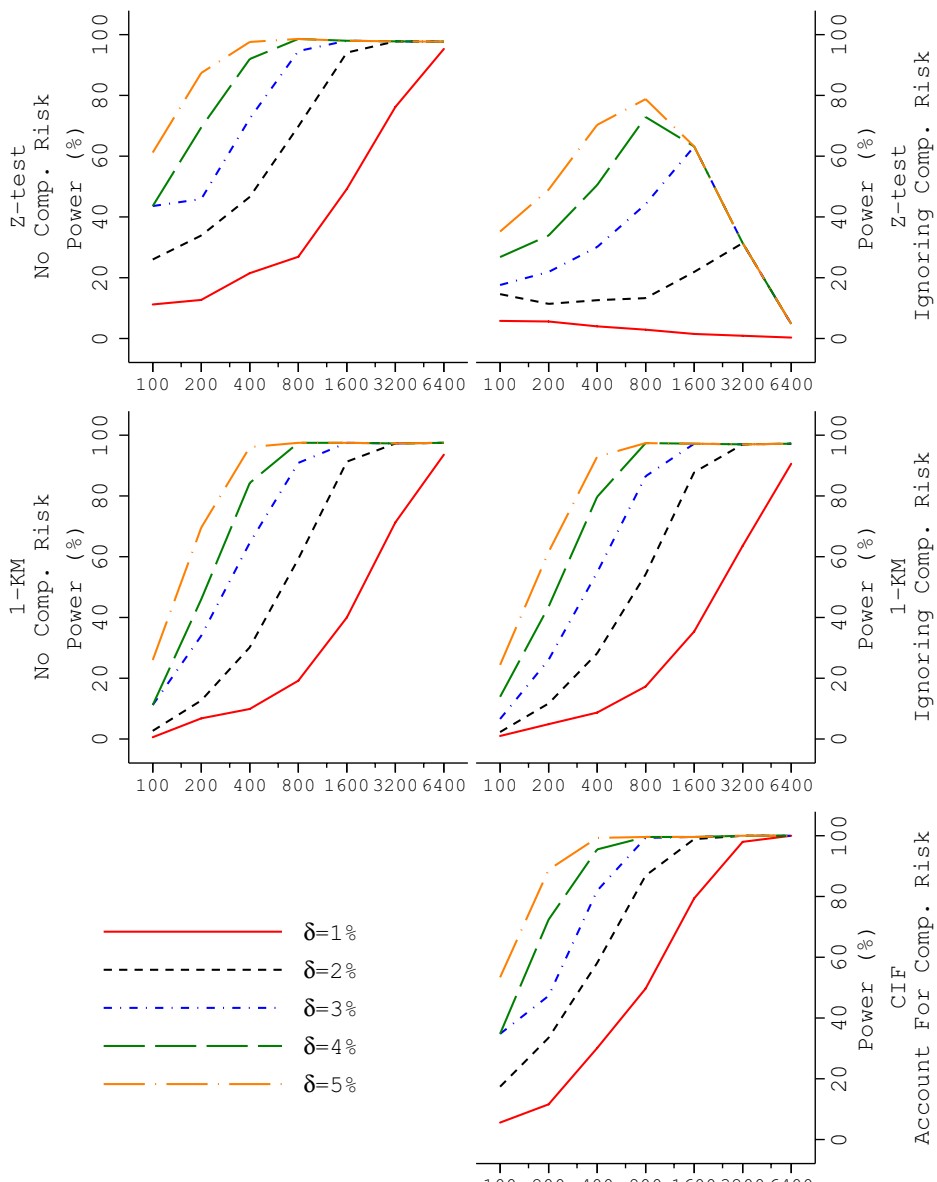

**Figure 5** Power to detect non-inferiority at 1%, 2%, 3%, 4% and 5% below a 95% benchmark performance. The data generating process and method of analysis are presented in separate panels. The sample size is indicated on the horizontal axis. CIF, cumulative incidence function.

Results from the simulation study using a non-inferiority paradigm are presented in figure 5.

When no CR is present, a 3% non-inferiority margin, and at samples sizes of 200 or 800, a z-test has 46% and 94% power to detect non-inferiority, respectively, and KM has 34% and 91% power to detect non-inferiority. When a non-informative CR is present, a 3% non-inferiority margin, and sample sizes n=200 or n=800, a z-test has 22% and 44% power to detect non-inferiority, respectively; 1−KM has 26% and 86% power to detect non-inferiority, respectively; and the CIF has 48% and 99% power to detect non-inferiority, respectively.

Mean performance estimates from the simulation study are tabulated in the online supplementary table 1.

## DISCUSSION

This study investigates how and how many individuals are required to demonstrate non-inferiority in the failure rate of a medical device (arthroplasty prosthesis) compared with an external benchmark in the presence or absence of a CR.

Net failure estimated using 1−KM provides unbiased estimates in the presence or absence of a non-informative

CR, whereas a simple z-test or CIF overestimate and underestimate failure in the presence of a non-informative CR, respectively. While there is reasonable agreement between analytical and simulation estimates of the sample size required to conduct a non-inferiority study, the failure to incorporate an adjustment for censoring due to mortality leads to erroneous estimates of power.

Using 1−KM to estimate failure in the presence of a non-informative CR (estimating net failure), a sample size of n=1600, will ensure coverage at nominal levels, have a CI width of approximately 2.3% and a RMSE of 0.46% but have a 35%, 88%, 97%, 97% and 97% power of demonstrating non-inferiority at margins of 1%, 2%, 3%, 4% and 5%, respectively.

This study has a number of strengths. We have shown the well-known differences between net and crude failure. Despite the persistent suggestion in arthroplasty research that 1−KM overestimates implant failure, it is clear that it is an unbiased estimate of net failure but a biased estimate of crude failure, a patient's personal chance of revision surgery. CIF (crude failure) in a CR model unsurprisingly underestimates net failure, assuming independence between CR and implant failure. We have demonstrated the number of individuals required at the onset of a benchmarking study for a variety of non-inferiority margins, under two different DGPs, and the width of the estimated CI. Despite the exemplar of arthroplasty, similar methods would apply to any discipline interested in conducting a formal benchmarking process.

However, this study has a number of important limitations. (1) We simulated data from a Weibull model with an uncorrelated CR, and while it provides a convenient and sensible method of generating data in an arthroplasty example, more complex models with correlated CRs may be appropriate in other areas. (2) We assume that time to revision surgery is a reasonable outcome of interest. However, alternative outcomes, for example, Patient Reported Outcome Measures, could similarly be incorporated into a non-inferiority benchmarking design. (3) The threshold for revision is assumed to be homogenous between different surgeons in this simulation. (4) We have not considered the effect of analysing data from multiple sources or combining pre-existing data. However, methods of meta-analysis in non-inferiority settings are well documented.[28 29] (5) The continuous accrual of data and regular assessment of data against benchmarks of interest will inevitably lead to multiple hypothesis testing, and therefore, the risk of incorrectly accepting or rejecting the null hypothesis of inferiority against the benchmark will increase. The decision to repeatedly look at the data must be made a priori, and due consideration for the multiple testing should be made in order to preserve the type I error rate.[30] Related to this point is that, once a prosthesis meets the benchmark level of performance, this should be considered the beginning of the monitoring process, and as new data are accrued, periodic assessment should occur to ensure that satisfactory performance is being maintained. (6)

Current implant rating organisations consider the failure rates of individual components (ie, head, stem and cups separately) opposed to the implanted construct. Despite the technical difficulties of disentangling the failures of each individual component, the multiple testing that ensues will ultimately require larger samples in order to preserve the type I error rate. (7) We have adopted a conservative scenario, where the implant failure rate is equal to the benchmark; in a more optimistic scenario, where the implant failure rate is less than the benchmark, fewer individuals will be required to detect non-inferiority. Conversely, when implant failure rate is more than the benchmark but less than the non-inferiority margin, larger samples will be required. However, unless implant manufacturers are, a priori, willing to propose superiority opposed to non-inferiority, a conservative standpoint would appear sensible. Finally and most importantly, (8) we critically assume that a simple unadjusted analysis provides a causal interpretation, and there is no residual confounding or selection bias that inhibits making a fair comparison. However, we recognise that assumption has little validity given the strong variation in implant use and failure rates between those young and old and between males and females.[24] Therefore, the necessity to adequately describe the population used within the benchmarking process or adopt a stratified approach to benchmarking is compelling. Similarly, the current design of arthroplasty benchmarking studies, specifically the use of an external benchmark, makes more sophisticated methods of adjusting data, via regression, weighting or matching, unfeasible. Therefore, there may be a number of benefits to choosing a contemporary prosthesis combination with established performance to which all other prosthesis can be benchmarked against. We also recognise as with nearly all clinical studies and registers that enrolled or consenting participants can be substantially different from those not enrolled or those who withhold consent to be included in a register. These potential differences may have a profound impact on how the results from benchmarking studies should be interpreted.

Despite some authors suggesting that a 1−KM overestimates implant survival in the presence of CR,[7] it is clear that this is not universally true, and it is very important to understand what is being estimated (net or crude failure) and for what purpose, recognise that implant failure rate is not necessarily equal to the rate of revision surgery in arthroplasty and ensure that analyses reflect the research question being presented.[31] The three methods of analysis, that is, a z-test (simple proportion), 1−KM and CIF, can provide equivalent or subtly different estimates depending on DGP. When data are fully complete (DGP1), that is, everyone is fully observed for the period of interest and there is no censoring, the failure rate estimated by a simple proportion, 1−KM and CIF will all be identical; variability in coverage and power will be due to inherent differences in the methods used to construct CIs. When there is censoring due to a CR (ie, DGP2), differences in the estimates arise.

These differences are best understood when data are fully observed for individuals experiencing the event of interest or the relevant competing event. A simple proportion (z-test) excludes patients who experience the competing event from the denominator, whereas CIF includes them; the numerator for both methods is simply the observed number of failures at 10 years. In this scenario, it is easy to see that failure rate estimated by a simple proportion will be greater than CIF due to the reduction in the denominator. It is also clear that CIF does not estimate the proportion of failures that would have occurred in those individuals who have experienced the competing event and therefore will underestimate the net failure rate of the device (the numerator is too small relative to the denominator), whereas, the 1−KM method excludes individuals from the analysis who experience CR, thereby reducing the denominator, which results in a higher failure rate than CIF.

This balancing act is very well known and demonstrated by the seminal work of Gooley et al.[32] They generally recommend that researchers report CIF but similarly note that, if one is interested in evaluating a cause-specific failure, CIF may be misleading, and inferences should be made from functions that are based solely on the hazard of failure from the cause of interest; that is, use the KM estimator. Putter et al[8] similarly state that the 'naive Kaplan-Meier estimator describes what would happen if the competing event could be prevented to occur, creating an imaginary world in which an individual remains at risk of failure from the event of interest,' that is, an immortal patient cohort. Ranstam et al[31] describe this in an arthroplasty setting as the 'implicit assumption that the patient will be alive until the implant fails.' This is not to say that CIF (crude failure) is not of interest to arthroplasty surgeons, patients, regulators or healthcare planners. It is just to say that CIF (crude failure) estimates a different quantity, which reflects the real-world (observed) probability of implant failure (arthroplasty revision rate) while acknowledging that CR of mortality explicitly prevents failure from the cause of interest. If CIF was used in a benchmarking setting, this may have a number of different consequences: (1) it could lead to a number of implants being claimed as non-inferior to the benchmark, when they are in fact inferior, due to differences between crude and net failure, and (2) the method of estimation may lead to the selective introduction of implants in older patients where mortality is higher, which is likely to reduce the reported failure rate.

The broad success of arthroplasty, as well as currently used benchmark failure rate of 5% at 10 years, necessitates the use of small non-inferiority margins. A small non-inferiority margin of 1% in absolute risk represents a 20% increase in relative risk of failure compared with a benchmark of 5%. The minimum numbers of individuals required to demonstrate non-inferiority of a device where its true failure rate is equal to the benchmark is unsurprisingly large. Modest sample sizes (n=1600) have limited power (35%) to detect non-inferiority, and only when sample sizes become large (n=6400) does power increase substantially (90%).

Despite differences in the performance of estimators and interpretation of estimands, the choice of sample size at the beginning of a study should be based on the desire to obtain sufficiently precise estimates (small RMSE and narrow CI's), which mitigates type II errors for a given non-inferiority margin, which is tolerable to the public, surgeons and regulators, whereas the choice of sample size that should remain at risk at the end of the benchmarking period is somewhat more difficult to determine. Two possible reasons to ensure the numbers at risk at the end of the period are large include (1) maintaining the performance and minimising the width of CIs and (2) ensuring that sufficient numbers of patients experience 10 years of the risk of revision for the estimates and the benchmarking process to be credible.

From a conservative perspective, it is simple to request that the same number of patients for a given level of power, $(1 - \beta)$, and non-inferiority margin, $\delta$, at the beginning of the benchmarking period, remain at risk at the end of the benchmarking period. However, this ignores that censoring due to mortality is expected to occur in an elderly populations and will delay the benchmarking process, whereas, it would not be difficult to request a specified delay (ie, 10 years) from surgery on the $j^{th}$ patient until the benchmarking assessment is made, a process that is similar to preregistration of randomised trials. This will implicitly allow for censoring due to mortality, without prespecifying how much censoring will occur, and assumes there is no loss to follow-up (which is often assumed in arthroplasty registers).

## CONCLUSION

The choice of non-inferiority margin, initial sample size or desired width of the CI in a benchmarking study are all subjective decisions and can only be chosen by balancing the risk of incorrectly awarding a benchmarking standard to an implant with a failure rate beyond the non-inferiority margin versus benchmark inflation where all devices receive benchmarks and the entire process lacks credibility. However, this study clearly demonstrates how 1−KM provides unbiased estimates of net implant failure, in a conservative scenario when the failure rate of an implant being tested is equal to the benchmark and has a CR that is uncorrelated to the event of interest.

**Contributors** AS conceived and designed the study. AS, MC, AJ, MW and AB interpreted the data, revised the manuscript and approved the final draft.

**Funding** AS was supported by a MRC strategic skills fellowship: MRC Fellowship MR/L01226X/1.

**Competing interests** None declared.

**Provenance and peer review** Not commissioned; externally peer reviewed.

**Data sharing statement** No data are available to be shared.

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
