## [Reviewer comments · BMJ Open]

ARTICLE DETAILS

TITLE (PROVISIONAL)	Determining the sample size required to establish whether a medical device is non-inferior to an external benchmark.
AUTHORS	Sayers, Adrian; Crowther, Michael; Judge, Andrew; Whitehouse, Michael; Blom, Ashley

VERSION 1 - REVIEW

REVIEWER	Jonas Ranstam, professor of medical statistics (retired) Dept of Clinical Sciences Lund, Orthopedics, Lund University, Sweden
REVIEW RETURNED	26-Feb-2017

GENERAL COMMENTS	This is a well-written manuscript describing important problems that arise when designing a study to compare the failure rate of a prosthetic implant with an external benchmark, especially with regard to sample size, trial design (superiority and non-inferiority), and estimands (net and crude failure rate). The results from a number of analytical calculations and simulations are presented, and this seems to have been well performed. I have just a few comments and questions. 1. The argument for using a non-inferiority trial design is that a superiority design is unsuitable when the implant's failure rate equals the benchmark. However, accepting a non-inferiority margin above the benchmark, which is necessary for the non-inferiority approach (5 different non-inferiority margins are included in the study) implies that the investigator considers a specified number of failures to be of no clinical importance. Even if the specified non-inferiority margin is small, a large number of patients can be affected. Would this not present an ethical dilemma?2. The EMA guideline on the choice of the non-inferiority margin (EMA/CPMP/EWP/2158/99) suggests running a superiority trial with a significance level greater than 0.05 as a possible alternative to using a non-inferiority margin. Could this be an alternative also when comparing an implant failure rate with a benchmark? What would the consequences be in terms of sample size?3. The simulation of implant failure and mortality have been based on two different Weibull distributions. How were the parameters of these distributions chosen? Were they estimated empirically or was just suitable fictitious values used?
--

	4. It is known that mortality rates, and perhaps also implant failure rates, differ between young and old and between men and women, which have consequences for the patterns of censoring and competing risk events. Age and sex have, however, not been directly addressed in the calculations and simulations. Does this have consequences for the practical applicability of the described sample sizes? Can we use them as they are, independent of the characteristics of the study population?
--	--

REVIEWER	SA Lie The Norwegian Arthroplasty Register, Norway
REVIEW RETURNED	08-Mar-2017

GENERAL COMMENTS	Determining the sample size required to establish whether a medical device is non-inferior to an external benchmark. This article discusses an interesting concept in how many implants that are needed to identify when a non-inferior conclusion can be drawn in an arthroplasty register on the basis on a benchmark. There is however some issues that needs to be clarified or discussed. In my opinion arthroplasty registries should be used to identify inferior implants and products. Aiming at identifying non-inferior implants has several problems. The authors state that "We assume simple analyses using Kaplan Meier derived from population registry studies can provide causal interpretations." This is a very strict assumption and is just briefly discussed! If the observed survival of implants is different, the first question would be if the risk set (age, gender, type of fixation, etc) for the two products are comparable. The same applies if the observed survival is equal! The authors should argue more elaborate on this - and maybe also discuss alternative approaches to handle this important issue. Can e.g. sub-sampling, propensity scores, propensity matching, or adjustment of the non-inferiority analyses be considered? For all situations the acquired sample size would increase. Plain unadjusted analyses (KM or CIFs) will rarely be sufficient to argue for the quality Another minor issue that can be raised is the continuous surveillance that arthroplasty registers offer. Hence, even if the analyses at some time show non-inferiority, the conclusion may differ with more data collected.
--

REVIEWER	Antti Eskelinen, MD, PhD Coxa Hospital for Joint Replacement, Tampere, Finland
REVIEW RETURNED	08-Mar-2017

GENERAL COMMENTS	General comments: This is a well-conducted simulation study assessing the
---

	methods and sample size to conduct a non-inferiority study of a medical device for the purposes of benchmarking. The research question is important, the methods seem to be valid (even though an expert statistical review is required), and the results are logically presented and adequately discussed. There are only some minor issues that I would like the authors to address. They can be seen in Specific Comments below. Specific Comments: Abstract: 1) This is clear and concise, nothing to revise here. Introduction 1) This is also well written, as the authors provide a very comprehensive background for this study. However, this section is a bit lengthy, and the reader would appreciate a more concise approach. Methods 1) In the abstract, the authors state the setting of this study as “National register of medical devices”. In the methods, however, they have not further described this issue. Even though this is a simulation study, have the authors somehow utilized real national register data to build up their database that they have used for simulations? Results 1) This is very well structured section, and the figures nicely support the text. Discussion 1) Page 13, limitations. They have listed “3) The threshold for revision is homogeneous between different surgeons” as one important limitation. As it is well known, that this is not the case in the real-world situation, I assume that the authors mean that this applies to this simulation setting. I’d like the authors to clarify this issue. 2) In this paper, the authors have very nicely shown that in a simulation setting, 1-KM provides unbiased estimates of net implant failure thus being a nice tool for benchmarking implant performance. However, there are many possible confounders in implant benchmarking. One of them is selection bias, i.e. there may large biological differences between the cohorts of patients treated with different implants. Thus, should these analyses be adjusted for age and sex (at least), for instance, by using Cox multiple regression model? Further, the smaller the samples are, the more likely comorbidities may affect the outcomes. For example, if morbid obesity combined with DM is significantly more prevalent in cohort of patients treated with knee replacement A than in patients receiving a knee replacement in general, this will most probably result in increased incidence of knee revisions for PJI in the cohort A and may also result in inferior survivorship of implant A as compared to benchmark at 10 years. This may also affect the required sample size. I’d like the authors to comment these issues.
--	---

REVIEWER	Obioha Ukoumunne University of Exeter Medical School, UK
REVIEW RETURNED	23-Mar-2017

GENERAL COMMENTS	This study presents the sample size required to establish whether a medical device is non-inferior to an established benchmark, based on a true failure rate of 5% for the medical device and the benchmark. The study also uses simulation compare the properties of different analytical methods for estimating failure rate in scenarios where there is: (a) no competing risk; (b) a competing risk (mortality). Major points  1. The paper addresses an important relevant topic but it was not clear what the unique contribution is. Methods for calculating the number of participants needed for non-inferiority tests for a single sample are established. One line in the abstract suggests that it is not currently clear how many implants are needed to establish non-inferiority, but isn't it more the case that the correct methods for calculating sample size are not currently being used? This paper still has value if the aim is to bring these methods (and required sample size for relevant scenarios) to the attention to an audience that is not already aware of their existence but it is not completely clear that this is the motivation for the paper. Related to this I wondered whether the paper is targeting non-statisticians/clinicians who are doing benchmark studies or targeting statisticians. Some parts of the paper look complex for the former group but many in the latter group would be aware of the issues that are being discussed. So I think the authors should be more direct about why this paper is unique. 2. Given that these methods for sample size and analysis are already available I wondered how researchers typically calculate sample size and analyse data from medical device performance studies. It would be good to give a few examples of papers where inappropriate methods have been used and demonstrate the importance of this paper for spreading the message. Are there any papers reporting benchmark studies that have used appropriate methods? The first paragraph in the Introduction section says there is no consensus but are some methods used more than others? 3. The paper sets out the importance of examining the problem under the non-inferiority framework but some of the simulations also examine performance of the methods under the superiority framework. Why is the superiority scenario considered of interest? 4. The paper indicates that the net failure rate is the quantity of interest but they examine the performance of the competing risk model that estimates the crude failure rate. It was not clear why they examine a method that inherently estimates a quantity that is different to the one of interest. I think the abstract and introduction sections should be explicit that the net failure rate is of interest. 5. Provide a justification for using 1000 replications for each simulation scenario – for example, how precisely does it
--

	enable you to estimate coverage of the 95% confidence interval. 6. In the “Performance” section provide the definitions (or calculations) for bias, root mean square error, power, confidence interval coverage and confidence interval width. 7. Is power a relevant concept in the context of superiority analyses when, as here, the failure rate of the medical device is the same as the benchmark? Isn't then just Type I error? 8. The results for both data generating scenarios are shown on the same figure (Figure 4) but this would be more readable if separate figures are used for each data generating scenario. Minor points  1. Is the term “1-Kaplan Meier” standard use? 2. The points made in the Conclusion section of the Abstract may well be true but they could have been written before the analyses were carried out; that is they do not directly result from the analyses in this paper. 3. Does anybody still carry out benchmark studies based on a failure rate of 10%, rather than 5%? How does this impact on the sample size requirement and performance of the analytical methods? 4. The Strengths and Limitations section (point 4) says that 5 analytical methods were examined but it is only 3; 2 methods in one scenario and all 3 methods in the other scenario. 5. The Abstract indicates the methods were assessed based on POWER, bias and precision but the Strengths and Limitations section lists bias, ROOT MEAN SQUARE ERROR, COVERAGE and confidence interval width. I'd make these lists consistent with each other. 6. In Figure 1 make it clearer what H0 means (i.e., null hypothesis) and that the footnotes are based on the non-inferiority framework rather than the superiority framework. 7. On the third page of the Introduction section there is a reference to 100% power. Is perfect power actually attainable? 8. On the third page of the Introduction section there is a reference to a hypothetical scenario where the upper confidence interval bound is always greater than the benchmark. While this would be true on average it wouldn't necessarily be the case in every single study as some estimates will be underestimates. 9. Provide a reference for the ADMEP structure for describing simulation studies. 10. Make it clearer in the “Method of Analysis” section that the failure function approach used in DGP 1 is the same as the “1-KM” approach used for analysis in DGP 2 if this is the case.
--	---

	11. The “Performance” section suggests that root mean square error is the same as “absolute bias”. I think this should be “absolute error” as some error is due to variability rather than bias? 12. One line in the results section implies that RMSE is a measure of variability. It is a measure of overall accuracy as it incorporates error due to both variability (variance) and bias. 13. One line in the results section says the “CIF ..consistently underestimates net failure by 0.5%” but Figure 4 makes this look as if it is the KM CR method that underestimates by this amount. 14. Most of the Figures are not numbered (e.g., Figure 4). 15. Ashley Blom’s first name is spelt incorrectly on the cover page.
--	---

VERSION 1 – AUTHOR RESPONSE

Reviewer: 1

Reviewer Name: Jonas Ranstam, professor of medical statistics (retired)

Institution and Country: Dept of Clinical Sciences Lund, Orthopedics, Lund University, Sweden

Please state any competing interests: None declared

Please leave your comments for the authors below

This is a well-written manuscript describing important problems that arise when designing a study to compare the failure rate of a prosthetic implant with an external benchmark, especially with regard to sample size, trial design (superiority and non-inferiority), and estimands (net and crude failure rate). The results from a number of analytical calculations and simulations are presented, and this seems to have been well performed. I have just a few comments and questions.

We are very grateful to Professor Rasntam for taking the time to read this manuscript and providing helpful commentary.

1. The argument for using a non-inferiority trial design is that a superiority design is unsuitable when the implant's failure rate equals the benchmark. However, accepting a non-inferiority margin above the benchmark, which is necessary for the non-inferiority approach (5 different non-inferiority margins are included in the study) implies that the investigator considers a specified number of failures to be of no clinical importance. Even if the specified non-inferiority margin is small, a large number of patients can be affected. Would this not present an ethical dilemma?

This is an excellent question, and you are indeed correct that the ethics of selecting a large or small non-inferiority margin are exceptionally important in this population.

Given the high prevalence of arthroplasty in the population, a 5% non-inferiority margin with a 5% benchmark at 10 years could represent a 100% increase in the failure rate of a prosthesis and a substantial clinical burden. Given the global use of prostheses the numbers of implants influenced could be very large.

We do not believe there is a simple answer to this question. But we hope that formalizing the sample sizes required in order to make non-inferiority statements will stimulate future debate within the orthopaedic community and relevant stakeholders which include patients.

We now cite guidance from the FDA and EMA in the introduction as they provide extensive commentary on choosing an appropriate non-inferiority margin.

Page 7 Para 2

2. The EMA guideline on the choice of the non-inferiority margin (EMA/CPMP/EWP/2158/99) suggests running a superiority trial with a significance level greater than 0.05 as a possible alternative to using a non-inferiority margin. Could this be an alternative also when comparing an implant failure rate with a benchmark? What would the consequences be in terms of sample size?

We thank you for directing us to this interesting discussion of an alternative approach to specifying non-inferiority margins.

As the EMA suggest, we re-structured the simulation to demonstrate superiority using less stringent confidence intervals (p-values). We varied this Confidence interval from 0 to 95% to illustrate the potential extremes.

The results suggest if the treatment of interest is exactly equal to the standard treatment (benchmark), a 0% confidence interval will give no more than 50% power to detect superiority regardless of the sample size. This is not that surprising considering the unbiased statistic will be symmetrically distributed around the truth i.e. 50% superior, and 50% inferior.

Therefore the utility of such a method is unclear given this will always be the case if the two treatments of interest are exactly equal.

Despite the three scenarios discussed by the EMA:

- 1. The products truly are equally efficacious, leaving a non-inferiority trial as the only option.**

2. The test product has a small advantage that would require such a large trial to detect as to be impractical.
3. The product has a disadvantage, but that disadvantage is smaller than a proposed non-inferiority margin.

It appears that scenario 1 and scenario 3 are not applicable.

Scenario 1 describes that the products being tested are truly efficacious, and the non-inferiority design as previously described is appropriate

Scenario 3, is somewhat niche, and we concur with the EMA that it would be better if products that had a disadvantage failed more times than they succeeded. Given its rather niche application we believe it is more likely to confuse people than illuminate. However, we note they will have the power to detect a difference that is less than 50% for all sample sizes, how much less will depend on the magnitude of the disadvantage and sampling variability.

Therefore scenario 2 is the only scenario where this approach could be considered sensible, and much like a normal superiority study, power will depend on the magnitude and variability of the advantage at a given sample size. But similarly, we feel this rather niche application is likely to confuse opposed to illuminate.

We include the results of this simulation for your interest.

- · — Z-test NCR
- - - Z-test CR
- — — KM NCR
- - - - KM CR
- · - · - CIF CR

3. The simulation of implant failure and mortality have been based on two different Weibull distributions. How were the parameters of these distributions chosen? Were they estimated empirically or was just suitable fictitious values used?

Suitable fictitious values which are close to those estimated from real data. We now expand on their choice in the methods.

Page 9 Para 4, & Page 10 Para1

4. It is known that mortality rates, and perhaps also implant failure rates, differ between young and old and between men and women, which have consequences for the patterns of censoring and competing risk events. Age and sex have, however, not been directly addressed in the calculations and simulations. Does this have consequences for the practical applicability of the described sample sizes? Can we use them as they are, independent of the characteristics of the study population?

This is an excellent point. We know the patterns of implant use and the rates of revision between young and old and men and women are very different. The applicability of one simple omnibus test statistic, therefore, might be considered unreasonable or of little practical value.

The requirement to conduct a stratified benchmarking approach may be more relevant with age and sex specific benchmarks, thus clearly emphasising the richness or generalisability of the data used. Furthermore, the requirement to adjust for multiple testing given the potentially large number of strata of interest is another important consideration. We have expanded our discussion to reflect this more fully.

Page 15 Para 1

Reviewer: 2

Reviewer Name: SA Lie

Institution and Country: The Norwegian Arthroplasty Register, Norway

Please state any competing interests: None declared

Please leave your comments for the authors below

Determining the sample size required to establish whether a medical device is non-inferior to an external benchmark.

This article discusses an interesting concept in how many implants that are needed to identify when a non-inferior conclusion can be drawn in an arthroplasty register on the basis on a benchmark.

We thank Professor Lie for taking the time to read this manuscript and provide helpful commentary.

There is however some issues that needs to be clarified or discussed.

In my opinion arthroplasty registries should be used to identify inferior implants and products. Aiming at identifying non-inferior implants has several problems. The authors state that “We assume simple analyses using Kaplan Meier derived from population registry studies can provide causal interpretations.” This is a very strict assumption and is just briefly discussed!

If the observed survival of implants is different, the first question would be if the risk set (age, gender, type of fixation, etc) for the two products are comparable.

The same applies if the observed survival is equal! The authors should argue more elaborate on this - and maybe also discuss alternative approaches to handle this important issue. Can e.g. sub-sampling, propensity scores, propensity matching, or adjustment of the non-inferiority analyses be considered? For all situations the acquired sample size would increase. Plain unadjusted analyses (KM or CIFs) will rarely be sufficient to argue for the quality

We have a great deal of sympathy with this perspective, and the ability to infer true non-inferiority from simple unadjusted analyses might be considered so limited that they are difficult to generalise to clinical practice.

However, the orthopaedic community is pioneering, rightly and potentially wrongly, the principles and approaches of benchmarking with relatively little discussion in the academic literature. The current system of benchmarking is widely accepted and broadly adopted despite its limitations. We believe defining a sufficiently large sample in order to make inferences is one aspect of benchmarking worthy of comment and that would act to evolve and improve the current system.

Another area also worthy of substantial comment is the suitability of a simple omnibus statistic to reflect true implant performance.

The principle problem of the effect of age and gender on implant failure rates in an external benchmarking context is the inability to adjust, as there is nothing with variability to compare it against, i.e. the benchmark is just a scalar.

The identification and use of a contemporary benchmark may alleviate many of these problems, as a proven prosthesis gives rise to the possibility of conducting a more meaningful comparison with potential for adjustment, whether that be by stratification, regression, matching, or inverse probability weighting.

We have now discussed this very relevant observation in more depth.

Page 15 Para1

Another minor issue that can be raised is the continuous surveillance that arthroplasty registers offer. Hence, even if the analyses at some time show non-inferiority, the conclusion may differ with more data collected.

This is another very relevant comment, and repeated early testing may produce highly variable results, therefore the decision to test early and benefit from chance low failure rates should be strongly discouraged. If products are benchmarked when sufficient data has been collected the variability in the estimates should be less. However, the necessity to perform continuous testing on products currently meeting the benchmarks is clear. We have now emphasised this point more strongly.

Page 15 Para 1

Reviewer: 3

Reviewer Name: Antti Eskelinen, MD, PhD

Institution and Country: Coxa Hospital for Joint Replacement, Tampere, Finland

Please state any competing interests: None declared

Please leave your comments for the authors below

General comments:

This is a well-conducted simulation study assessing the methods and sample size to conduct a non-inferiority study of a medical device for the purposes of benchmarking. The research question is important, the methods seem to be valid (even though an expert statistical review is required), and the results are logically presented and adequately discussed. There are only some minor issues that I would like the authors to address. They can be seen in Specific Comments below.

We thank Professor Eskelinen for taking the time to read this manuscript and providing very helpful and constructive comments.

Specific Comments:

Abstract:

1) This is clear and concise, nothing to revise here.

Introduction

1) This is also well written, as the authors provide a very comprehensive background for this study. However, this section is a bit lengthy, and the reader would appreciate a more concise approach.

We have attempted to provide a more structured approach to the introduction so the more statistically aware individuals can skip the basic introduction of non-inferiority designs.

Page 5 new subsection “Hypothesis tests and power”

Page 6 new subsection “Non-inferiority and benchmarking studies”

Page 7 new subsection “Aims”

Methods

1) In the abstract, the authors state the setting of this study as “National register of medical devices”. In the methods, however, they have not further described this issue. Even though this is a simulation study, have the authors somehow utilized real national register data to build up their database that they have used for simulations?

We have now changed the abstract to indicate the setting is a “Simulation study of a National Register of Medical Devices” to make it clear that data is purely simulated.

Page 1 Para 4

Results

1) This is very well structured section, and the figures nicely support the text.

We thank you for this positive comment

Discussion

1) Page 13, limitations. They have listed “3) The threshold for revision is homogeneous between different surgeons” as one important limitation. As it is well known, that this is not the case in the real-world situation, I assume that the authors mean that this applies to this simulation setting. I’d like the authors to clarify this issue.

This is correct, we assume in the simulation that the decision to revise a prosthesis is homogenous across all procedures and surgeons. In the real world, we know this is not the case, and variability due to the decision to revise will only attenuate power, and require larger samples. We have revised the text to indicate this an implicit assumption in the simulation.

Page 14 Para 5

2) In this paper, the authors have very nicely shown that in a simulation setting, 1-KM provides unbiased estimates of net implant failure thus being a nice tool for benchmarking implant performance. However, there are many possible confounders in implant benchmarking. One of them

is selection bias, i.e. there may large biological differences between the cohorts of patients treated with different implants. Thus, should these analyses be adjusted for age and sex (at least), for instance, by using Cox multiple regression model? Further, the smaller the samples are, the more likely comorbidities may affect the outcomes. For example, if morbid obesity combined with DM is significantly more prevalent in cohort of patients treated with knee replacement A than in patients receiving a knee replacement in general, this will most probably result in increased incidence of knee revisions for PJI in the cohort A and may also result in inferior survivorship of implant A as compared to benchmark at 10 years. This may also affect the required sample size. I'd like the authors to comment these issues.

These are all excellent comments, and we wholeheartedly agree with your observations. The necessity to adjust for at least age and gender is clear, and conducting sex and age specific benchmarks would be one method of addressing this issue. However, more complex adjustment is limited by the design i.e. an external benchmark. The use of a contemporary prosthesis with proven performance could potentially serve as a more relevant, contemporaneous and useful benchmark which others can be compared against.

Furthermore, we agree that with a small and highly selected sample that certain co-morbidities may be more or less prevalent than in other cohorts. In this instance, we concur that the observed failure rate may be influenced by co-morbidities of the patients in addition to the performance of the prosthesis.

We have now reflected more fully on this problem in the discussion.

Page 15 Para 1

Reviewer: 4

Reviewer Name: Obioha Ukoumunne

Institution and Country: University of Exeter Medical School, UK

Please state any competing interests: None declared

Please leave your comments for the authors below

This study presents the sample size required to establish whether a medical device is non-inferior to an established benchmark, based on a true failure rate of 5% for the medical device and the benchmark. The study also uses simulation compare the properties of different analytical methods for estimating failure rate in scenarios where there is: (a) no competing risk; (b) a competing risk (mortality).

We thank Professor Ukoumunne for taking the time to read and provide helpful comments on this manuscript.

Major points

1. The paper addresses an important relevant topic but it was not clear what the unique contribution is. Methods for calculating the number of participants needed for non-inferiority tests for a single sample are established.

We agree that methods of estimating sample size in a one sample non-inferiority study of a single proportion are available. However, we do not believe this has been investigated in a survival analysis setting, where censoring is present. We now make this clear in our aims.

Page 7 Para 4

One line in the abstract suggests that it is not currently clear how many implants are needed to establish non-inferiority, but isn't it more the case that the correct methods for calculating sample size are not currently being used?

We believe we are correct in our assertion. Simple methods that do not account for censoring are available. However, when censoring is present this is not the case, we are not aware of any studies investigating this. Therefore we believe this statement is accurate.

This paper still has value if the aim is to bring these methods (and required sample size for relevant scenarios) to the attention to an audience that is not already aware of their existence but it is not completely clear that this is the motivation for the paper.

We thank you for recognising the value of this study, and raising awareness of these issues in this context is one motivation for the paper. But, we still believe the application of a simulation study to investigate a one-sample non-inferiority study design of failure against an external benchmark is both novel and of relevance.

Related to this I wondered whether the paper is targeting non-statisticians/ clinicians who are doing benchmark studies or targeting statisticians. Some parts of the paper look complex for the former group but many in the latter group would be aware of the issues that are being discussed. So I think the authors should be more direct about why this paper is unique.

We hope the manuscript has broad appeal. Whilst the introduction of hypothesis testing and non-inferiority testing will be repetitive for statisticians and quantitative epidemiologists, we hope that it provides context for those less familiar with study design. We believe that one of the problems with current broadly accepted processes is different groups (e.g. statisticians/clinicians) using different information or criteria to define the assumptions used to determine what is an acceptable or unacceptable implant, hence our desire to produce a paper on the issues we have identified that has sufficient information to be of use but remains accessible to experts from different backgrounds.

We hope the more detailed discussion of estimands and performance metrics to appeal more strongly to the more quantitatively able readers.

Therefore we feel the paper will have dual appeal, to this ends we have added sub-sections to the introduction to direct the readers and allow the more statistically adept to skip them easily.

We have now revised the manuscript to make this clear.

Page 5 new subsection “Hypothesis tests and power”

Page 6 new subsection “Non-inferiority and benchmarking studies”

Page 7 new subsection “Aims”

2. Given that these methods for sample size and analysis are already available I wondered how researchers typically calculate sample size and analyse data from medical device performance studies. It would be good to give a few examples of papers where inappropriate methods have been used and demonstrate the importance of this paper for spreading the message. Are there any papers reporting benchmark studies that have used appropriate methods? The first paragraph in the Introduction section says there is no consensus but are some methods used more than others?

The introduction of arthroplasty prostheses into the market do not routinely require product benchmarking or randomised clinical trials. Within the European Union a simple CE accreditation, demonstrating “equivalence” to an existing product is all that is required. Therefore the literature to provide examples does not exist. We are currently aware of three groups currently performing product benchmarking, all listed in the methods, and we do not believe it is constructive to provide a detailed critique of each group's methodology, whereas we would prefer to highlight some of the important assumptions required in a benchmarking study which we hope they may adopt.

3. The paper sets out the importance of examining the problem under the non-inferiority framework but some of the simulations also examine performance of the methods under the superiority framework. Why is the superiority scenario considered of interest?

The superiority framework is, in essence, a non-inferiority design with a non-inferiority margin at zero. It would seem incomplete to not include this in the analysis and demonstrates why a non-inferiority design should be adopted.

We now specify this in the performance section of the methods.

Page 11 Para 3

4. The paper indicates that the net failure rate is the quantity of interest but they examine the performance of the competing risk model that estimates the crude failure rate. It was not clear why they examine a method that inherently estimates a quantity that is different to the one of interest. I think the abstract and introduction sections should be explicit that the net failure rate is of interest.

In the abstract, we now specify that net failure estimated using 1-KM is of specific interest and preferential to crude failure estimated using competing risks model.

Page 2, para 7 & 8

In the introduction, we specify that there is ambiguity with regards what the appropriate estimand is, and therefore we feel that illustrating the differences to net and crude failure emphasises the importance of picking the appropriate estimand.

5. Provide a justification for using 1000 replications for each simulation scenario – for example, how precisely does it enable you to estimate coverage of the 95% confidence interval.

1000 replications were sufficiently large to be perceived plausible, and also large enough to provide small Monte Carlo error. Assuming appropriate coverage of the test statistics in a binomial proportion, the Monte Carlo error is $(0.95(1-0.95)/n)^{1/2}=0.007$ or 0.7%. We now specify this rationale.

Page 10 Para 1

6. In the “Performance” section provide the definitions (or calculations) for bias, root means square error, power, confidence interval coverage and confidence interval width.

We have now included definitions

Page 11 Para 3

7. Is power a relevant concept in the context of superiority analyses when, as here, the failure rate of the medical device is the same as the benchmark? Isn't this then just Type I error?

Power to detect superiority in a non-inferiority study design is equivalent to setting a non-inferiority margin at zero. Whilst this is almost equivalent to type I error rate, power in this context dictates that only the upper bound confidence interval is less than the benchmark. If *a priori* you specified a one-tail test, they would indeed be equivalent. We have updated the methods to reflect this more clearly.

Page 11 Para 3

8. The results for both data generating scenarios are shown on the same figure (Figure 4) but this would be more readable if separate figures are used for each data generating scenario.

We thank you for this suggestion, but we feel that it then becomes more difficult to compare the two different scenarios, and it becomes harder to compare the relative performance metrics of the different methods. Given the comments of reviewer 3 in favour of the current figures, we have kept the current figure 4.

Minor points

1. Is the term "1-Kaplan Meier" standard use?

Unfortunately much of the Orthopaedic literature just simply refer to results from a Kaplan-Meier analysis as KM, even if it is actually 1-KM. Therefore, for the sake of specificity and clarity we use 1-KM, so the language is familiar, but also specify in the methods the failure function.

2. The points made in the Conclusion section of the Abstract may well be true but they could have been written before the analyses were carried out; that is they do not directly result from the analyses in this paper.

We have now restructured the conclusions of the abstract to reflect the results of the simulation study.

3.

Does anybody still carry out benchmark studies based on a failure rate of 10%, rather than 5%? How does this impact on the sample size requirement and performance of the analytical methods?

We believe some groups are currently still benchmarking hip prostheses at 10% at 10 years. However, the performance of the best performing implant within the NJR of England and Wales is closer to 2.5% at 10 years, therefore the applicability of calculating benchmarks at 10% at 10 years is unclear.

4. The Strengths and Limitations section (point 4) says that 5 analytical methods were examined but it is only 3; 2 methods in one scenario and all 3 methods in the other scenario.

This is correct, we have amended the section to make this explicit.

Page 3 para 1

5. The Abstract indicates the methods were assessed based on POWER, bias and precision but the

Strengths and Limitations section lists bias, ROOT MEAN SQUARE ERROR, COVERAGE and confidence interval width. I'd make these lists consistent with each other.

We have now made them consistent.

Page 1 Para 6

6. In Figure 1 make it clearer what H0 means (i.e., null hypothesis) and that the footnotes are based on the non-inferiority framework rather than the superiority framework.

We have now clarified that H0 is the null hypothesis, and that footnote relates to the non-inferiority framework.

See Figure 1

7. On the third page of the Introduction section there is a reference to 100% power. Is perfect power actually attainable?

Perfect power is probably not attainable, but the reference to 100% power emphasises the interpretation of what power means. We now indicate this is likely impossible.

Page 6 Para 2

8. On the third page of the Introduction section there is a reference to a hypothetical scenario where the upper confidence interval bound is always greater than the benchmark. While this would be true on average it wouldn't necessarily be the case in every single study as some estimates will be underestimates.

Your interpretation is correct. However, we do say in the sentence immediately prior to this, that if the estimated failure is exactly equal to the benchmark. This predicates the comment with regards to non-zero confidence intervals. We now link the two sentences more clearly.

Page 7 Para 1

9. Provide a reference for the ADMEP structure for describing simulation studies.

ADMEP is a structure described by Tim Morris, Ian White, and Michael Crowther at a simulation course. We believe this first explicit published use of ADMEP. We now attribute ADMEP to Morris, White, and Crowther.

Page 9 Para 1

10. Make it clearer in the “Method of Analysis” section that the failure function approach used in DGP 1 is the same as the “1-KM” approach used for analysis in DGP 2 if this is the case.

We have now use the same terminology to make this clearer.

Page 10 Para 3 & 4

11. The “Performance” section suggests that root mean square error is the same as “absolute bias”. I think this should be “absolute error” as some error is due to variability rather than bias?

This is correct, we have now corrected this.

Page 11 Para 3

12. One line in the results section implies that RMSE is a measure of variability. It is a measure of overall accuracy as it incorporates error due to both variability (variance) and bias.

We agree and have now updated this.

Page 12 Para 3

13. One line in the results section says the “CIF ..consistently underestimates net failure by 0.5%” but Figure 4 makes this look as if it is the KM CR method that underestimates by this amount.

We apologise for the incorrect legend, a typographical error in our code led to the incorrect line style being used. We have now corrected this.

14. Most of the Figures are not numbered (e.g., Figure 4).

We have now included figure titles and numbers in the manuscript. We assume the numbering will occur in the production of the manuscript.

Page 19

15. Ashley Blom’s first name is spelt incorrectly on the cover page.

Thank you for spotting this error, we have amended his name.

Page 1

VERSION 2 – REVIEW

REVIEWER	Dr Jonas Ranstam Lund University, Sweden
REVIEW RETURNED	02-Jun-2017

The reviewer completed the checklist but made no further comments.

REVIEWER	Antti Eskelinen Coxa Hospital for Joint Replacement, Tampere, Finland
REVIEW RETURNED	02-Jun-2017

GENERAL COMMENTS	I'm satisfied with the authors' response and the changes they've incorporated into the revised manuscript.
--

REVIEWER	Obioha Ukoumunne University of Exeter, UK
REVIEW RETURNED	10-Jun-2017

GENERAL COMMENTS	I am happy with the revised manuscript.
---